# Tetracycline Resistance Genes in Wild Birds from a Wildlife Recovery Centre in Central Italy

**DOI:** 10.3390/ani13010076

**Published:** 2022-12-24

**Authors:** Antonietta Di Francesco, Daniela Salvatore, Fabrizio Bertelloni, Valentina Virginia Ebani

**Affiliations:** 1Department of Veterinary Medical Sciences, University of Bologna, 40064 Ozzano dell’Emilia, Italy; 2Department of Veterinary Sciences, University of Pisa, Viale delle Piagge 2, 56124 Pisa, Italy

**Keywords:** wildlife, free-living birds, tetracycline resistance genes, PCR

## Abstract

**Simple Summary:**

Antimicrobial resistance (AMR) is recognised as an urgent global threat, both in human and in veterinary medicine. In recent years, increasing interest has turned to wild animals in order to evaluate their role as additional sources of antimicrobial resistance. Resistant bacteria may be cycling through wildlife and back into the ecosystem, then wildlife populations may act as sentinels for AMR in the environment. In this study, an epidemiological investigation of the spread of tetracycline resistance (*tet*) genes in wild birds was performed using a culture-independent approach. Positivity for one or more *tet* genes was found in 114 (45%) of 254 free-living birds tested. In view of the growing anthropogenic pressure, the spread of antimicrobial resistance in wildlife and the implications for resistance control strategies will have to be considered carefully.

**Abstract:**

Wild animals are less likely to be exposed directly to clinical antimicrobial agents than domestic animals or humans, but they can acquire antimicrobial-resistant bacteria through contact with humans, animals, and the environment. In the present study, 254 dead free-living birds belonging to 23 bird species were examined by PCR for the presence of tetracycline resistance (*tet*) genes. A fragment of the spleen was collected from each bird carcass. A portion of the intestine was also taken from 73 of the 254 carcasses. Extracted DNA was subjected to PCR amplification targeting the *tet*(L), *tet*(M), and *tet*(X) genes. In total, 114 (45%) of the 254 birds sampled belonging to 17 (74%) of the 23 bird species tested were positive for one or more *tet* genes. The *tet*(M) gene showed a higher frequency than the other tested genes, both in the spleen and in the intestine samples. These results confirm the potential role of wild birds as reservoirs, dispersers, or bioindicators of antimicrobial resistance in the environment.

## 1. Introduction

Antimicrobial resistance (AMR) is a complex phenomenon involving three interdependent ecosystems of potential relevance to public health: humans, animals, and the natural environment. Antimicrobial resistance naturally occurs due to the production of antimicrobial molecules by strains of bacteria and fungi in all environments, including soil [1,2]. However, in the last decades, there has been an acceleration caused by the misuse and overuse of antimicrobials in both human and veterinary medicine, as well as in livestock, where antimicrobials have been used as growth supplements [3]. In the veterinary field, most studies on AMR have focused on the animal species most subjected to pharmacological pressure, such as intensively reared cattle, pigs, poultry, and fish, implicated as reservoirs for multidrug-resistant foodborne pathogens. Usually, wild animals are less likely to be exposed directly to clinical antimicrobial agents than domestic animals or humans [4], except occasionally in rehabilitation facilities. However, wildlife is part of the natural environmental compartment and can be influenced by anthropogenic pressures, e.g., by human waste systems or animal husbandry facilities that can be a source of active antimicrobials, antimicrobial-resistant bacteria, and resistance genes [5].

Conventional antimicrobial susceptibility testing methods are based on bacteriological culture and antimicrobial susceptibility testing of the isolated microorganisms. A limitation of culture-dependent methods is the existence of labile or viable but not culturable bacteria, as well microorganisms that require a long period of growth. Recent studies introduced molecular approaches based on amplification of antimicrobial resistance target genes from environmental or biological samples [6,7,8,9,10]. This approach is more expensive than traditional cultivation and does not allow determination of the bacterial sources of resistance genes. However, it is a fast method and avoids possible underestimation of the occurrence of AMR due to a consistent not culturable or labile fraction of microorganisms [11]. Moreover, considering AMR genes as contamination markers, methods which allow searching for these genes rather than for the bacteria carrying them could aid epidemiological efforts to analyse the spread of resistance determinants [12,13,14].

The aim of this study was to investigate by PCR the presence of tetracycline (*tet*) resistance genes in free-living birds in Italy.

## 2. Materials and Methods

### 2.1. Sampling

From 2016 to 2020, 254 free-living birds, belonging to 23 bird species, that died from trauma or predation (Table 1) were collected at a wildlife recovery centre located in Tuscany (Central Italy) and transported to the Department of Veterinary Sciences of Pisa University (Pisa, Italy) for educational activities. In the present study, only intact organs were examined. A fragment of the spleen was collected from each bird carcass and immediately stored at −20 °C. A portion of the intestine was also taken from 73 of the 254 carcasses.

### 2.2. Molecular Analysis

#### 2.2.1. DNA Extraction

Total DNA was individually extracted from each sample using the QIAamp DNA mini kit (Qiagen, Hilden, Germany), following the manufacturer’s instructions. Negative (kit reagents only) controls were included in each set of extraction. The DNA extracts were stored at −20 °C before analysis.

#### 2.2.2. DNA Amplification and Sequencing

DNA samples were investigated to search for three genes involved in the three tetracycline resistance mechanisms: *tet*(L) (tetracycline efflux pumps), *tet*(M) (ribosomal protection) and *tet*(X) (enzymatic inactivation). Each *tet* gene was amplified in an individual PCR, using primers described by Ng et al. [15] (Table 2).

Different PCR protocols were carried out: 5 min of initial denaturation at 94 °C followed by 35 cycles at 94 °C for 1 min; 53 °C (*tet*(M) and *tet*(X)) or 55 °C (*tet*(L)) for 1 min; and 72 °C for 1 min. A final step of 10 min at 72 °C completed the reaction. The DNA extracted from *Escherichia coli* field strains, containing tetracycline resistance plasmids, was used as a positive control. The extraction control and a distilled water negative control were also included.

The PCR products were analysed by gel electrophoresis (2% agarose); the DNA bands were stained with Midori Green Advance (Nippon Genetics Europe GmbH, Düren, Germany) and then visualised using ultraviolet (UV) trans illumination. The amplicons were purified using the High Pure PCR Product Purification Kit (Roche, Mannheim, Germany), and both DNA strands were sequenced (Bio-Fab Research, Rome, Italy). The sequences obtained were compared with the public sequences available using the BLAST server in GenBank (National Center for Biotechnology Information 2022).

## 3. Results

In total, of the 254 birds sampled, 114 (45%), belonging to 17 (74%) of the 23 bird species tested, were positive for one or more *tet* genes. With respect to the 254 tested spleens, 82 (32%) were *tet*(M) positive, 7 (3%) were *tet*(L) positive, 4 (1.5%) were *tet*(X) positive and 21 (8%) were positive for both *tet*(M) and *tet*(L) (Figure 1; Table 3).

With respect to the intestine samples, 18 (25%) of 73 samples collected were positive for one or more *tet* genes. In particular, 14 (19%) were positive for *tet*(M), 2 (3%) for *tet*(L) and 2 (3%) for both *tet*(M) and *tet*(L). No positivity for *tet*(X) was found in intestine samples (Table 4).

Each *tet* positive intestine sample was from a carcass whose spleen was positive for one or more *tet* genes. A total *tet* gene match was found between the spleen and intestine samples from heron, mallard, kestrel, and gull. A different *tet* gene distribution was found between the spleen and intestine samples from a Eurasian teal, showing a *tet*(L) positive intestine sample vs. a *tet* (M) positive spleen. For each *tet* gene amplified, the identity of the amplicon was confirmed by comparison between the sequence obtained and the corresponding sequences from antibiotic-resistant Gram-positive or Gram-negative bacteria in the GenBank database, showing 99–100% nucleotide similarity. One sequence for each of the three tetracycline resistance genes amplified was deposited in the GenBank database under accession numbers OP793879, OP793880, and OP807021.

## 4. Discussion

Microbes, antimicrobial agents, and AMR genes may be cycled and re-cycled through soil, groundwater, marine water, wild animals, crops, shellfish, and livestock [16]. Moreover, the expansion of urban populations and the changes in land use resulting in a fragmentation or loss of natural habitats can lead to the overcoming of natural barriers between humans and wildlife, increasing direct and indirect contact of wildlife with humans and their livestock and potentially expanding the role of wildlife in AMR propagation. Therefore, increasing interest has turned to wild animals in order to evaluate their role as an additional source of antimicrobial resistance.

Antimicrobial resistance has been reported in a wide range of wild animals, such as wild boars [17], wild rabbits [18], wolves [19], lynxes [20], red foxes [21], roe deer [4,22], wild rodents [23], European hedgehogs, and crested porcupines [9].

Wild birds have been speculated to be sentinels, reservoirs, and potential vehicles of resistant bacteria and genetic determinants of AMR [4,8,24,25,26]. In particular, migratory bird species traveling great distances in short periods of time and inhabiting a wide variety of environments could act as efficient AMR dispersers [2,27]. A role for migratory wild birds has also been proposed to explain the occurrence of multidrug-resistant bacteria in places that are isolated from human activities [27].

A characterisation of resistant bacteria isolated from wild birds showed a variety of antimicrobial resistance patterns, including tetracycline resistance [5,10,28,29,30,31,32,33,34].

The extensive use of tetracyclines in clinical practice and their incorporation into livestock feed at subtherapeutic doses as growth promoters, until 2006 when this practice was stopped in the EU [35], has subjected bacterial populations to selection pressure and increased the prevalence of tetracycline resistance, which is considered one of the most abundant antimicrobial resistance mechanisms among pathogenic and commensal microorganisms [36]. Tetracycline resistance is generally caused by the acquisition of tetracycline resistance (*tet*) genes, often associated with either a mobile plasmid or a transposon that act as vectors, transferring genetic information between bacteria of the same or different species [2,37]. To date, at least 59 *tet* genes and 11 mosaic *tet* genes have been described [38]. Three main resistance mechanisms are mediated by *tet* genes: pumping the drug out of the cell before it reaches its site of action (active efflux pumps), protection of the ribosomal binding site, which decreases drug binding, and enzymatic inactivation of the active compound. The first two mechanisms currently predominate in clinical settings [39].

In this study, we focused on three *tet* genes representative and frequently detected within each of the three mechanisms of resistance.

In total, 114 (45%) of 254 spleens tested showed one or more *tet* genes. In particular, *tet*(M) was detected in 103 (40.5%), *tet*(L) in 28 (11%), and *tet*(X) in 4 (1.5%) of the 254 spleen samples. With respect to the 73 intestine samples, 16 (22%) were *tet*(M) positive, and 4 (5%) were *tet*(L) positive. No intestine samples tested positive for *tet*(X). *Tet*(M) is one of 12 different *tet* genes that code for proteins that protect bacterial ribosomes from the action of tetracyclines. In particular, it has been proposed that the Tet(M) protein alters the conformation of the tetracycline binding site [40]. The high *tet*(M) frequency detected in this study, compared to that of the other *tet* genes tested, was consistent with other reports showing that this gene has the broadest taxonomic distribution among all bacteria, probably because of its association with conjugative chromosomal elements [39]. Conjugative transposons appear to have lower host specificity than plasmids, which may explain the detection of *tet*(M) in 81 different genera, including 40 Gram-positive and 41 Gram-negative bacteria [41,42]. *Tet* (L) is one of 33 *tet* genes that code for energy-dependent efflux of tetracyclines, i.e., the active transport of tetracyclines across the cell membrane. This gene is generally found on small transmissible plasmids. *Tet*(L) is one of the *tet* genes that in recent years has shown the largest increase in its distribution among bacterial genera. To date, it has been detected in 25 Gram-positive and 23 Gram-negative bacteria [41,42]. *Tet*(X) is one of the 13 genes encoding for enzymes that chemically modify tetracycline. In particular, the *tet*(X) gene encodes a NADP-dependent monooxygenase that catalyses the degradation of tetracycline antimicrobials [43], including tigecycline [44], a next-generation tetracycline that is considered as a last resort antimicrobial against severe infections by pan-drug-resistant bacterial pathogens [45]. This gene was not only observed in obligate anaerobes but also in a variety of environmental aerobic bacteria, as well as diverse human pathogens [45]. Until now, *tet*(X) has only been found in Gram-negative genera (n. 25), except the genus *Clostridium* [41,42].

The gene frequency observed in this study agreed with previous reports [41,42]. Tetracycline resistance genes are not randomly distributed among bacteria, but they occur with a different frequency depending the bacterial species and/or genera [46], directing the choice of the target genes on the basis of the bacterial isolates obtained. Otherwise, if culture-independent methods are applied, it would be advisable to test a panel of genes including those with the highest frequency, to avoid the risk of underestimation.

The high *tet* gene prevalence observed in magpies (71%) and hooded crows (62.5%) is not surprising, considering that these omnivorous and opportunistic species, as well as pigeons and herons, developed a marked synanthropic temperament, taking advantage of the presence of crops, waste, landfills, and all that derives from human activities.

The results observed in waterfowl species, including migratory species, and birds of prey agree with previous studies suggesting contamination via water sources where wastewater from farms and cities is discharged [47,48] or via their prey, such as small wild animals and birds with diverse habitat ranges, including urban and rural environments [49,50]. In this regard, Kozak et al. [51] reported that small wild mammals living in the proximity of farms are generally more likely to harbour resistant bacteria than wild mammals living in natural areas. Predatory birds can also feed on carcasses of livestock animals that can carry more antimicrobial resistant bacterial strains [33]. From an epidemiological perspective, the detection of resistance genes in migratory bird species, as well as in prey birds foraging across large distances, is of particular interest regarding their potential ability to disperse resistance genes or antimicrobial-resistant bacteria over large areas [14], accelerating the globalization of AMR [52].

Antimicrobial resistance genes are a normal finding in the commensal gut microbiota [27]. Considering that the molecular approach did not allow us to associate *tet* genes with bacterial species, the detection of *tet* genes from the intestine did not allow us to distinguish pathogenic from commensal bacterial source, whereas the amplification of *tet* genes from the spleen could be more suggestive of their association with bacteria responsible for infection in the animals tested. Contamination could be excluded because only intact intestines were examined, and DNA extraction was performed by removing an internal fragment from each organ with a disposable scalpel blade. Moreover, nine birds showing *tet* positive spleens had *tet* negative intestines.

Previous studies reported the presence of antimicrobial-resistant bacteria and resistance genes in wildlife admitted to rehabilitation centres [30,53,54,55]. Baros Jorquera et al. [5], focusing on a wildlife rehabilitation centre built environment as a possible source of antimicrobial-resistant bacteria in hospitalised animals, showed bacterial environmental contamination that did not entirely overlap with the antimicrobial resistance patterns detected in admitted animals. In this study, the tested birds had been collected already dead or dying and then passed quickly through the rehabilitation centre, where they received no antimicrobial treatment. Therefore, it is likely that they carried resistant bacteria before their admission to the centre. Workers in wildlife centres and people who may have professional or other contact with wild birds should consider the risk of transmission of zoonotic antimicrobial-resistant agents or transmissible resistance determinants and take appropriate preventive measures.

## 5. Conclusions

The acquisition of resistance genes in wild and free-ranging populations is a concern, as this may increase the environmental reservoir of AMR. The results of this study agree with previous reports suggesting the role of wildlife and wild birds in particular as reservoirs, vectors, or bioindicators of resistant bacteria and genetic determinants of antimicrobial resistance in the environment. In the future, this role could become more significant as wildlife, livestock, and people are more frequently in close contact due increased anthropogenic pressure.

## Figures and Tables

**Figure 1 animals-13-00076-f001:**
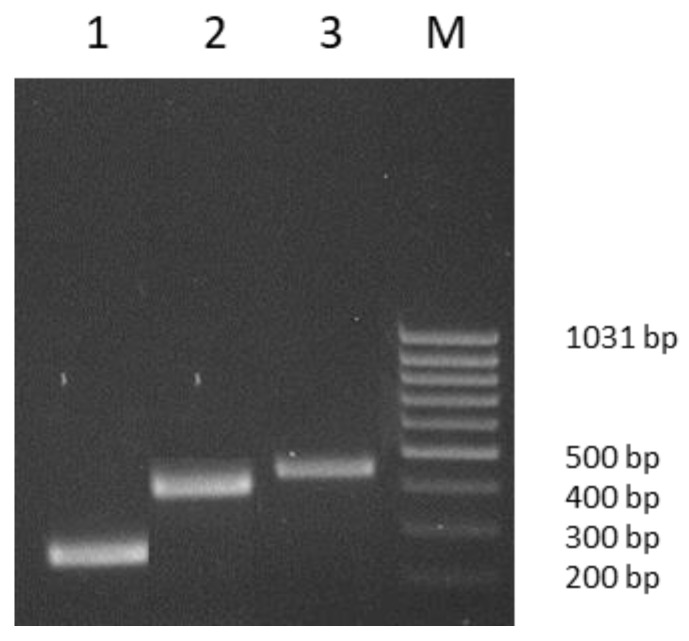
PCR amplicons. Lane 1, 267 bp *tet*(L) gene fragment; lane 2, 406 bp *tet*(M) gene fragment; lane 3, 468 bp *tet*(X) gene fragment; lane M, MassRuler Low Range DNA Ladder, (Thermo Fisher Scientific, Vilnius, Lithuania).

**Table 1 animals-13-00076-t001:** Bird species sampled.

Family	Bird Species	Spleen n.	Intestine n.
**Corvidae**	*Pica pica* (Eurasian magpie)	45	0
	*Corvus cornix* (hooded crow)	24	0
	**Subtotal**	**69**	**0**
**Ardeidae**	*Ardea cinerea* (heron)	2	2
	**Subtotal**	**2**	**2**
**Scolopacidae**	*Gallinago gallinago* (snipe)	5	0
	**Subtotal**	**5**	**0**
**Anatidae**	*Mareca strepera* (gadwall)	2	0
	*Anas acuta* (pintail duck)	4	1
	*Mareca penelope* (Eurasian wigeon)	23	11
	*Anas platyrhynchos* (mallard)	26	12
	*Spatula querquedula* (garganey)	1	0
	*Spatula clypeata* (shoveler duck)	10	2
	*Aythya fuligula* (tufted duck)	1	0
	*Aythya ferina* (pochard)	1	0
	*Anser anser* (goose)	1	0
	*Tadorna tadorna* (shelduck)	4	3
	*Anas crecca* (Eurasian teal)	79	22
	**Subtotal**	**152**	**51**
**Strigidae**	*Athene noctua* (owl)	1	1
	**Subtotal**	**1**	**1**
**Accipitridae**	*Accipiter nisus* (Eurasian sparrowhawk)	4	1
	**Subtotal**	**4**	**1**
**Falconidae**	*Falco peregrinus* (falcon)	1	1
	*Falco tinnunculus* (kestrel)	3	3
	**Subtotal**	**4**	**4**
**Rallidae**	*Fulica* (coot)	2	1
	**Subtotal**	**2**	**1**
**Columbidae**	*Columba palumbus* (wood pigeon)	3	1
	*Columba livia* (pigeon)	2	2
	**Subtotal**	**5**	**3**
**Laridae**	*Larus marinus* (gull)	10	10
	**Subtotal**	**10**	**10**
**TOTAL**		**254**	**73**

**Table 2 animals-13-00076-t002:** Primers used for the detection of tetracycline resistance target genes.

Primers	Sequence (5′-3′)	Target Gene	PCR Product (bp)
tetL-F	TCGTTAGCGTGCTGTCATTC	*tet*(L)	267
tetL-R	GTATCCCACCAATGTAGCCG	*tet*(L)	267
tetM-F	GTGGACAAAGGTACAACGAG	*tet*(M)	406
tetM-R	CGGTAAAGTTCGTCACACAC	*tet*(M)	406
tetX-F	CAATAATTGGTGGTGGACC	*tet*(X)	468
tetX-R	TTCTTACCTTGG CATCC G	*tet*(X)	468

**Table 3 animals-13-00076-t003:** Tetracycline resistance genes detected in the spleens of wild birds.

				Spleen		
Family	Bird Species	Sample n.	*tet*(L)	*tet*(M)	*tet*(L+M)	*tet*(X)
**Corvidae**	magpie	45	0	22 (49%)	10 (22%)	0
	hooded crow	24	0	13 (54%)	3 (12.5%)	0
	**Subtotal**	**69**	**0**	**35 (51%)**	**13 (19%)**	**0**
**Ardeidae**	heron	2	0	0	2 (100%)	0
	**Subtotal**	**2**	**0**	**0**	**2 (100%)**	**0**
**Scolopacidae**	snipe	5	0	0	0	0
	**Subtotal**	**5**	**0**	**0**	**0**	**0**
**Anatidae**	gadwall	2	0	1 (50%)	0	0
	pintail duck	4	0	2 (50%)	0	0
	Eurasian wigeon	23	0	4 (17%)	0	0
	mallard	26	1 (4%)	5 (19%)	1 (4%)	0
	garganey	1	0	0	0	0
	shoveler duck	10	0	1 (10%)	0	0
	tufted duck	1	0	0	0	0
	pochard	1	0	1 (100%)	0	0
	goose	1	0	0	0	0
	shelduck	4	0	1(25%)	0	0
	Eurasian teal	79	0	21 (26.5%)	2 (2.5%)	4 (5%)
	**Subtotal**	**152**	**1 (0.6%)**	**36 (24%)**	**3 (2%)**	**4 (3%)**
**Strigidae**	owl	1	0	0	1 (100%)	0
	**Subtotal**	**1**	**0**	**0**	**1 (100%)**	**0**
**Accipitridae**	Eurasian sparrow hawk	4	1 (25%)	2 (50%)	0	0
	**Subtotal**	**4**	**1 (25%)**	**2 (50%)**	**0**	**0**
**Falconidae**	falcon	1	1 (100%)	0	0	0
	kestrel	3	0	1 (33%)	2 (67%)	0
	**Subtotal**	**4**	**1 (25%)**	**1 (25%)**	**2 (50%)**	**0**
**Rallidae**	coot	2	0	0	0	0
	**Subtotal**	**2**	**0**	**0**	**0**	**0**
**Columbidae**	wood pigeon	3	0	2 (67%)	0	0
	pigeon	2	0	0	0	0
	**Subtotal**	**5**	**0**	**2 (40%)**	**0**	**0**
**Laridae**	gull	10	4 (40%)	6 (60%)	0	0
	**Subtotal**	**10**	**4 (40%)**	**6 (60%)**	**0**	**0**
**TOTAL**		**254**	**7 (3%)**	**82 (32%)**	**21 (8%)**	**4 (1.5%)**

**Table 4 animals-13-00076-t004:** Tetracycline resistance genes detected in the intestines of wild birds.

				Intestine		
Family	Bird Species	Sample n.	*tet*(L)	*tet*(M)	*tet*(L+M)	*tet*(X)
**Ardeidae**	heron	2	0	1 (50%)	0	0
	**Subtotal**	**2**	**0**	**1 (50%)**	**0**	**0**
**Anatidae**	pintail duck	1	0	0	0	0
	Eurasian wigeon	11	0	0	0	0
	mallard	12	0	3 (25%)	0	0
	shoveler duck	2	0	0	0	0
	shelduck	3	0	0	0	0
	Eurasian teal	22	1 (4.5%)	6 (27%)	0	0
	**Subtotal**	**51**	**1 (2%)**	**9 (18%)**	**0**	**0**
**Strigidae**	owl	1	0	0	0	0
	**Subtotal**	**1**	**0**	**0**	**0**	**0**
**Accipitridae**	Eurasian sparrow hawk	1	0	0	0	0
	**Subtotal**	**1**	**0**	**0**	**0**	**0**
**Falconidae**	falcon	1	0	0	0	0
	kestrel	3	0	0	2 (67%)	0
	**Subtotal**	**4**	**0**	**0**	**2 (50%)**	**0**
**Rallidae**	coot	1	0	0	0	0
	**Subtotal**	**1**	**0**	**0**	**0**	**0**
**Columbidae**	wood pigeon	1	0	0	0	0
	pigeon	2	0	0	0	0
	**Subtotal**	**3**	0	0	0	0
**Laridae**	gull	10	1 (10%)	4 (40%)	0	0
	**Subtotal**	**10**	**1 (10%)**	**4 (40%)**	**0**	**0**
**TOTAL**		**73**	**2 (3%)**	**14 (19%)**	**2 (3%)**	**0**

## Data Availability

The sequences generated in this study are available in GenBank under Accession numbers OP793879, OP793880, and OP807021.

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
