# Peer review of "Tetracycline Resistance Genes in Wild Birds from a Wildlife Recovery Centre in Central Italy"

_animals, 2022, doi:10.3390/ani13010076_

Round 1

Reviewer 1 Report

The paper entitled “Tetracycline resistance genes in wild birds in Italy” by Di Francesco et al., reports a screening of the prevalence of tet genes in wild birds collected by a rescue center in Italy over 5 years.

 The paper is well written and easy to read. However I am concerned whether this is enough for a publication. The authors extracted DNA from the spleen and (not for all the birds) the intestine. They use a culture-independent method (not an independent-culture method, as in line 14) to identify positive animals.

The authors amplify the 3 genes from a sample that has a complex microbial population: where the sequences all the same? What is the genetic variability of the tet genes? It seems strange to me that all the tet genes you isolated from such a high number of animals have one unique variant of each of these genes. If this is not the case, could you reconstruct phylogenies of different tet genes from different species and hypothezise where these genes come from?

What is the area of activity of the rescue center? Some of the birds you sample are migratory birds: I would expect to have a variability of these genes (but again, I don’t work directly with tet genes, so I am not sure what is the variability there).

Overall, culture-independent methods are coupled with NGS methods. I am surprised you managed to get “clean sequence” with total DNA extracted from spleen. Is it possible that across the 5 years of sampling you found these 3 genes in only one variation, with no SPNs?

Some minor comments:

Line 16: it is culture-independent approach.

Line 57: viable but not-culturable

Line 72: and transported to

Keep it consistent with “tet(x)” no space between tet and the brackets

Author Response

Dear Reviewer,

Thank you for your suggestions.

About your remarks:

The paper entitled “Tetracycline resistance genes in wild birds in Italy” by Di Francesco et al., reports a screening of the prevalence of tet genes in wild birds collected by a rescue center in Italy over 5 years.

The authors amplify the 3 genes from a sample that has a complex microbial population: where the sequences all the same? What is the genetic variability of the tet genes? It seems strange to me that all the tet genes you isolated from such a high number of animals have one unique variant of each of these genes. If this is not the case, could you reconstruct phylogenies of different tet genes from different species and hypothezise where these genes come from?

What is the area of activity of the rescue center? Some of the birds you sample are migratory birds: I would expect to have a variability of these genes (but again, I don’t work directly with tet genes, so I am not sure what is the variability there).

Overall, culture-independent methods are coupled with NGS methods. I am surprised you managed to get “clean sequence” with total DNA extracted from spleen. Is it possible that across the 5 years of sampling you found these 3 genes in only one variation, with no SPNs?

Regarding the variability of the tet genes, the literature does not describe a variability of the single genes. Roberts and Schwartz (Tetracycline and phenicol resistance genes and mechanisms: importance for agriculture, the environment, and humans. J. Environ. Qual. 2016, 45, 576−592) reported that “there is a clear link between bacterial taxonomy and specific types of ARGs, a phenomenon that has been particularly well documented with tetracycline resistance”. So, specific ARGs are found in only some bacterial species, but, to my knowledge, a variability of the nucleotide sequence of tet genes has not been described, neither between bacterial species nor between host species.

Furthermore, the regions amplified by the primers used in this study have been shown to be highly conserved, as demonstrated by the BLAST analysis of our amplicons:

-tet(L): 100% nucleotide identity (query cover 100%) with 100 sequences from 10 bacterial genera;

-tet(M): 100% nucleotide identity (query cover 100%) with 100 sequences from 14 bacterial genera;

-tet(X): 100% nucleotide identity (query cover 100%) with 100 sequences from 15 bacterial genera.

Similar results had been obtained by us in two previous publications concerning other animal species and other substrates:

-Di Francesco, A.; Salvatore, D.; Sakhria, S.; Catelli, E.; Lupini, C.; Abbassi, M.S.; Bessoussa, G.; Ben Yahia, S.; Ben Chehida, N. High Frequency and Diversity of Tetracycline Resistance Genes in the Microbiota of Broiler Chickens in Tunisia. Animals 2021, 11, 377.

-Antonietta Di Francesco, Maria Renzi, Nicole Borel, Hanna Marti, and Daniela Salvatore. Detection of Tetracycline Resistance Genes in European Hedgehogs (Erinaceus europaeus) and Crested Porcupines (Hystrix cristata). Journal of Wildlife Diseases, 56(1), 2020, pp. 219–223.

On the other hand, the purpose of this study was not to trace the antibiotic-resistant bacterial species, but to propose a culture-independent approach which is enjoying growing interest for epidemiological investigations on the spread of antimicrobial resistance.

The area of activity of the rescue center has been added.

Some minor comments:

Line 16: it is culture-independent approach.

The text has been modified.

Line 57: viable but not-culturable

The text has been modified.

Line 72: and transported to

The text has been modified.

Keep it consistent with “tet(x)” no space between tet and the brackets

The text has been modified.

Reviewer 2 Report

The manuscript presents the screening of wild bird carcasses for the presence of three tetracycline resistance genes by PCR. Authors findings highlight how wild bird could represent a reservoir of tetracycline resistant bacteria.

Major comments

1) A major limitation of the study is the number of the gene screened by PCR, given that only three tetracycline resistant genes, namely tet(L), tet(M) and tet(X) were tested. It is unclear to me why these genes were selected. In addition, although all PCR products were sequenced, no phylogenetic analysis was performed on gene sequences. This analysis, which is mainly relevant for tet(M), could be performed to investigate 1) the similarity between tet genes from the same bird (spleen vs. intestine), 2) similarity among the same sample type, and 3) similarity with published available sequences from known Gram-negative and Gram-positive species.

2) The title refers to wild birds in Italy and is too generic given that bird carcasses were collected from a single wildlife recovery center (line 72), for which the location has not been reported. Please correct.

3) Results are limited to 18 lines of manuscript and the main body of the paper is the introduction, which could be shortened, and the discussion which lacks a critical view on the data presented and would benefit of an original elaboration of the results, more than a repetition of them. For instance, a discussion on the consumption (human and veterinary use) of tetracycline in the area where these wild birds were collected would be interesting. Authors should also discuss why only the spleen and intestine where sampled. Some sentences are too speculative and not supported by results included in this study (e.g., lines 150-153, 189-196, 213-216).

Minor comments

1) Lines 41-43. Please add a reference to support your statement.

2) Line 56. Please clarify how “easily inactivated” should refer to bacteria.

3) Lines 60-61. Culture independent approaches are less expensive compared to traditional cultivation methods, which include isolation and susceptibility testing of bacteria.

4) The term “antimicrobials” and “antibiotics” are used interchangeably. Please uniform with “antimicrobials”.

5) Please correct “gram” with “Gram” were appropriate.

6) Line 166. Remove “do”.

7) Line 172. Please correct “gram-negative bacteria” with “Gram-negative genera”.

8) Lines 175-176. Please change “that is the final line of defense” with “that is considered as a last resort antimicrobial”.

8) The sentence on lines 179-180 is out of context. Please revise it or delete it.

Author Response

Dear Reviewer,

thank you for your remarks that helped improve the paper.

About your remarks:

Major comments

  • A major limitation of the study is the number of the gene screened by PCR, given that only three tetracycline resistant genes, namely tet(L), tet(M) and tet(X) were tested. It is unclear to me why these genes were selected. In addition, although all PCR products were sequenced, no phylogenetic analysis was performed on gene sequences. This analysis, which is mainly relevant for tet(M), could be performed to investigate 1) the similarity between tet genes from the same bird (spleen intestine), 2) similarity among the same sample type, and 3) similarity with published available sequences from known Gram-negative and Gram-positive species.

 “Only three tetracycline resistant genes were tested”.

Below are some examples of studies that have investigated a limited number of tet genes:

-Blanco-Peña, K.; Esperón, F.; Torres-Mejía, A.M.; de la Torre, A.; de la Cruz, E.; Jiménez-Soto, M. Antimicrobial resistance genes in pigeons from public parks in Costa Rica. Zoonoses Public Health 2017, 64, e23–e30: two tet [(tet(A) and tet(Q)] genes tested.

-Silva N, Igrejas G, Figueiredo N, Gonçalves A, Radhouani H, Rodrigues J, Poeta P. Molecular characterization of antimicrobial resistance in enterococci and Escherichia coli isolates from European wild rabbit (Oryctolagus cuniculus). Sci Total Environ. 2010 Sep 15;408(20):4871-6: two tet [(tet(A) and tet(B)] genes tested in resistant E. coli isolates and two tet [(tet(L) and tet(M)] genes tested in resistant enterococci isolates.

-Rogers SW, Shaffer CE, Langen TA, Jahne M, Welsh R. Antibiotic-Resistant Genes and Pathogens Shed by Wild Deer Correlate with Land Application of Residuals. Ecohealth. 2018 Jun;15(2):409-425: one tet [tet(Q)] gene tested.

- Gambino D, Vicari D, Vitale M, Schirò G, Mira F, Giglia M, Riccardi A, Gentile A, Giardina S, Carrozzo A, Cumbo V, Lastra A, Gargano V. Study on Bacteria Isolates and Antimicrobial Resistance in Wildlife in Sicily, Southern Italy. Microorganisms. 2021 Jan 19;9(1):203: one tet [tet(A)] gene tested.

- Dolejska, M.; Cizek, A.; Literak, I. High prevalence of antimicrobial-resistant genes and integrons in Escherichia coli isolates from black-headed gulls in the Czech Republic. J. Appl. Microbiol. 2007, 103, 11−19: two tet [tet(A) and tet(B)] genes tested.

  1. Guenther, S.; Grobbel, M.; Lübke-Becker, A.; Goedecke, A.; Friedrich, N.D.; Wieler, L.H.; Ewers, C. Antimicrobial resistance profiles of Escherichia coli from common European wild bird species. Vet. Microbiol. 2010, 29, 219−225: three tet [tet(A), tet(B), and tet(C)] tested.
  2. Kozak, G.K.; Boerlin, P.; Janecko, N.; Reid-Smith, R.J.; Jardine, C. Antimicrobial resistance in Escherichia coli isolates from swine and wild small mammals in the proximity of swine farms and in natural environments in Ontario, Canada. Appl. Environ. Microbiol. 2009, 75, 559−566: three tet [tet(A), tet(B), and tet(C)] tested.

“It is unclear to me why these genes were selected”.

As explained in the text (lines 78-80 and 160-161), we focused on three genes representative of the three mechanisms of action of tetracyclines [tet(L)-tetracycline efflux pumps, tet(M)-ribosomal protection, and tet(X) enzymatic inactivation]. Based on the literature, these genes were frequently detected within each of the three mechanisms of resistance.

Phylogenetic analysis would have been unproductive because the amplified regions were highly conserved, as demonstrated by the BLAST analysis of our amplicons:

-tet(L): 100% nucleotide identity (query cover 100%) with 100 sequences from 10 bacterial genera;

-tet(M): 100% nucleotide identity (query cover 100%) with 100 sequences from 14 bacterial genera;

-tet(X): 100% nucleotide identity (query cover 100%) with 100 sequences from 15 bacterial genera.

2) The title refers to wild birds in Italy and is too generic given that bird carcasses were collected from a single wildlife recovery center (line 72), for which the location has not been reported. Please correct.

The title has been modified.

  • Results are limited to 18 lines of manuscript and the main body of the paper is the introduction, which could be shortened, and the discussion which lacks a critical view on the data presented and would benefit of an original elaboration of the results, more than a repetition of them. For instance, a discussion on the consumption (human and veterinary use) of tetracycline in the area where these wild birds were collected would be interesting. Authors should also discuss why only the spleen and intestine where sampled. Some sentences are too speculative and not supported by results included in this study (e.g., lines 150-153, 189-196, 213-216).

“…..the main body of the paper is the introduction, which could be shortened…”

The text has been modified.

A discussion on the consumption of tetracycline in the area where the wild birds were collected would be inappropriate as most of the bird species sampled were migratory or prey birds.

Considering that the birds examined had died from traumatic or predatory events, only the intact organs were examined. The text has been modified (line 67).

Sentence lines 150-153: the text has been modified.

Sentences lines 189-196: considering that we also examined migratory and prey birds, these sentences, supported by bibliographic references, do not seem speculative or inappropriate.

Maybe I didn't understand your comment.

Sentences 213-216: in this sentence we meant that the genes highlighted in this study could hardly be linked to antimicrobial resistant bacteria present in the recovery center.

Minor comments

  • Lines 41-43. Please add a reference to support your statement.

A reference has been added (Plaza-Rodríguez, C., Alt, K., Grobbel, M., Hammerl, J.A., Irrgang, A., Szabo, I., Stingl, K., Schuh, E., Wiehle, L., Pfefferkorn, B.; et al. Wildlife as sentinels of antimicrobial resistance in Germany? Front. Vet. Sci. 2021, 7, 627821).

2) Line 56. Please clarify how “easily inactivated” should refer to bacteria.

The text has been modified.

3) Lines 60-61. Culture independent approaches are less expensive compared to traditional cultivation methods, which include isolation and susceptibility testing of bacteria.

Molecular investigations are more expensive than culture, in terms of the cost of equipment and reagents and of staff training. Galhano et al. (Antimicrobial resistance gene detection methods for bacteria in animal-based foods: a brief review of highlights and advantages. Microorganisms 2021, 9, 923) in Table 1 indicated the low cost among the advantages of the traditional method compared to molecular methods.

Molecular 4) The term “antimicrobials” and “antibiotics” are used interchangeably. Please uniform with “antimicrobials”.

The text has been standardized.

5) Please correct “gram” with “Gram” were appropriate.

The text has been modified.

6) Line 166. Remove “do”.

The text has been modified.

7) Line 172. Please correct “gram-negative bacteria” with “Gram-negative genera”.

The text has been modified.

8) Lines 175-176. Please change “that is the final line of defense” with “that is considered as a last resort antimicrobial”.

The text has been modified.

8) The sentence on lines 179-180 is out of context. Please revise it or delete it.

The sentence has been deleted.

Reviewer 3 Report

Dear authors, the article contains valuable information in the study of antimicrobial resistance. I have some observations

The title is confusing, as it is written it indicates that the resistance genes are from wild birds and according to the methodology they are resistance genes present in Gram positive and negative bacteria.The title and objective must be modified.

Writing the results is very basically  and there is no statistical analysis, there are several wild birds species, perhaps a comparative analysis can be made.

The authors must place a representative image of the PCR

Check throughout the document that it is correct Gram positive and Gram negative.

Author Response

Dear Reviewer,

thank you for your suggestions.

About your remarks:

The title is confusing, as it is written it indicates that the resistance genes are from wild birds and according to the methodology they are resistance genes present in Gram positive and negative bacteria. The title and objective must be modified.

Our molecular approach does not allow to trace which bacterial species the genes belong to. We can only state that tet genes were detected in the spleen or intestine of the sampled birds, such as in this reference: Blanco-Peña, K.; Esperón, F.; Torres-Mejía, A.M.; de la Torre, A.; de la Cruz, E.; Jiménez-Soto, M. Antimicrobial resistance genes in pigeons from public parks in Costa Rica. Zoonoses Public Health 2017, 64, e23–e30.

Writing the results is very basically and there is no statistical analysis, there are several wild birds species, perhaps a comparative analysis can be made.

Considering the great numerical inhomogeneity of the tested species, we did not consider it appropriate to perform a statistical evaluation.

The authors must place a representative image of the PCR

An image of the amplicons obtained has been included in the text.

Check throughout the document that it is correct Gram positive and Gram negative.

The text has been checked and modified.

Round 2

Reviewer 2 Report

Authors have addressed all my comments.

Reviewer 3 Report

The authors had to consider a representative number of samples per species that would allow them to carry out comparisons and statistical inferences between the different species..